# Associations between Milk Intake and Sleep Disorders in Chinese Adults: A Cross-Sectional Study

**DOI:** 10.3390/nu15184079

**Published:** 2023-09-21

**Authors:** Jinzhong Xu, Jiaying Lao, Qingxi Jiang, Wenhui Lin, Xiyi Chen, Chongrong Zhu, Shencong He, Wenbo Xie, Fan Wang, Bo Yang, Yanlong Liu

**Affiliations:** 1Department of Clinical Pharmacy, Affiliated Wenling Hospital, Wenzhou Medical University, Wenling 317500, China; fengzhongjin@163.com; 2School of Mental Health, Wenzhou Medical University, Wenzhou 325035, China; y935414587@163.com (J.L.); 18767739071@163.com (X.C.); 18705795165@163.com (C.Z.); hsc_20040511@163.com (S.H.); fttuwb@163.com (W.X.); 3Department of Preventive Medicine, School of Public Health and Management, Wenzhou Medical University, Wenzhou 325035, China; jqx990819@163.com; 4Cardiovascular Medicine, Affiliated Wenling Hospital, Wenzhou Medical University, Wenling 317500, China; lwh2163@126.com; 5Beijing Hui-Long-Guan Hospital, Peking University, Beijing 100096, China; fanwang@bjmu.edu.cn; 6Zhejiang Provincial Clinical Research Center for Mental Disorders, The Affiliated Wenzhou Kangning Hospital, Wenzhou Medical University, Wenzhou 325000, China

**Keywords:** milk intake, Pittsburgh Sleep Quality Index, sleep disorders, sleep disturbances, triglycerides

## Abstract

We aimed to examine the association of milk intake with sleep disorders and their specific indicators. The current study included 768 adults aged 28–95 from Wenling, China. Milk intake was assessed using a food frequency questionnaire with ten food items, while sleep disorders were measured using the Pittsburgh Sleep Quality Index (PSQI), with higher scores indicating poorer sleep. The participants were divided into two groups according to the average intake of milk per week: rare intake (≤62.5 mL/week) and regular intake (>62.5 mL/week). Primary measurements were multivariate-adjusted odds ratios (ORs) with 95% confidence intervals (CIs) for the prevalence of sleep disorders concerning regular milk intake compared with rare intake. In secondary analyses, linear regression analyses were performed to assess the effects of milk intake on sleep disorders and their specific dimensions. Regular intake of milk did not have a significant association with sleep disorders compared with rare intake (adjusted OR: 0.72, 95%; CI: 0.51, 1.03), but this association was found to be pronounced with sleep disturbances (OR: 0.49, 95%; CI: 0.28, 0.87). Increased intake of milk was significantly associated with the lower scores of PSQI for sleep quality (β: −0.045, 95%; CI: −0.083, −0.007) and sleep disturbances (β: −0.059, 95%; CI: −0.090, −0.029), respectively. When stratified by age and gender, the benefits of milk intake for sleep disorders and sleep disturbances were more significant in older adults (≥65) and men than in younger persons and women. In summary, regular milk intake benefits sleep quality, which may contribute to nutritional psychiatric support for prevention against sleep disorders.

## 1. Introduction

Sleep disorders are a global epidemic, affecting 45% of the world’s population. Sleep disorders increase the risk of physical diseases such as type 2 diabetes and hypertension and mental disorders such as depression and self-harm [1,2]. Current treatment options include non-pharmacological and pharmacological treatments (including melatonin, antipsychotic medication, antidepressants, α−adrenergic agonists, and sedative-hypnotic drugs) [3]. It is worth noting that the impact of dietary factors, including milk on sleep disorders, have received substantial attention [4,5,6].

Substantial evidence from population-based and animal studies supports a beneficial association between milk consumption and sleep disorders [2]. Clinical studies and animal studies suggested that functional foods that promote human sleep include vegetables such as lettuce [7], fruits such as cherries and kiwi [8,9], and milk [6]. A cross-sectional study involving 19,892 people found that consuming milk and dairy products before bedtime effectively improves sleep [2]. A recent cohort study found that higher milk consumption in adolescents was associated with earlier sleep timing (i.e., milk intake promoted early bedtime) [5].

There is physiological evidence linking milk intake to sleep disorders. Laboratory data have found melatonin, tryptophan, and biologically active peptides in milk that are essential in promoting sleep quality [10,11]. Melatonin is a sleep/wake hormone that plays a critical role in the onset and continuity of sleep [11]. Tryptophan is the precursor of the neurotransmitter serotonin and melatonin for sleep/wake [9,11]. In addition, biologically active peptides act on GABAergic neurons to regulate sleep [12,13].

Previous studies have examined the potential benefits of milk intake on sleep quality [14,15,16]. However, the association between specific indicators of sleep disorders and milk intake in Chinese adults are not fully understood. Therefore, we hypothesized that milk would improve sleep disorders, as well as specific indicators of sleep disorders. The present study has two aims: (1) to determine the association between milk and sleep disorders and (2) to identify the specific indicators of sleep disorders that milk affects.

## 2. Materials and Methods

### 2.1. Study Design

The present study is a cross-sectional analysis that aims to assess the association between dietary patterns, lifestyle behaviors, and psychosomatic disorders among the population residing in the Taizhou city of Zhejiang Province, China. General patients were recruited from Wenling Hospital affiliated with the Wenzhou Medical University between March 2018 and August 2019. Trained investigators completed a detailed questionnaire including items regarding socio-demographic characteristics, lifestyle variables, history of illnesses, and medication use in a face-to-face interview with the participants. Laboratory results and special examination results of the patients were collected.

### 2.2. Participants

The study participants were enrolled at baseline within an ongoing cohort in Zhejiang Province, China. An original plan for the cohort was to recruit 3000 general patients at baseline. The first phase plan was to recruit 1500 persons by posting posters at Wenling Hospital, affiliated with the Wenzhou Medical University, Zhejiang Province between March 2018 and August 2019. Ultimately, a total of 1106 general patients aged 25–95 years residing in Wenling City were enrolled in this study, with a response rate of 73.73%. The theoretical sample size of this study was set at 1056 individuals to provide a specific relative precision of 2% (Type I error, 0.05; Type II error, 0.10), taking into account an anticipated 80% participation rate. Epidemiology survey data included blood-based biochemical parameters, anthropometric measurements, socio-demographics, dietary intake, and lifestyle characteristics. We excluded 338 subjects for the following reasons: (i) lack of demographic and clinical disease information (*n* = 241); (ii) lack of food frequency questionnaire or PSQI data (*n* = 220); (iii) lack of blood samples (*n* = 45); and (iiii) have history of cow’s milk proteins intolerance (*n* = 0). The final study sample comprised 768 participants (mean age 64.71  ±  10.51 years, male 64.32%). For detailed information on participant recruitment, please refer to Figure 1. All participants provided written informed consent before data collection. The ethics committee of the Affiliated Wenling Hospital approved this study.

### 2.3. Clinical and Biochemical Measurements

All anthropometric and demographic data were measured by trained medical staff using standard protocols. Demographic data included age, gender, height, and weight. Body mass index (BMI) was calculated by dividing the participant’s weight (kg) by height (meter) squared. The trained investigators obtained historical information using a standard questionnaire, including alcohol consumption, smoking, past medical history (hypertension, diabetes, and coronary heart disease), and related medication use. Depression and anxiety states were assessed using the Hospital Anxiety and Depression Scale, with a score of ≥8 indicating depressive and anxious states. Blood samples were fasting morning samples collected in serum separator tubes. Samples were left at room temperature for 30 min and then centrifuged at 4000 r/min for 10 min to isolate serum. Serum alanine aminotransferase (ALT), aspartate aminotransferase (AST), lactate dehydrogenase (LDH), fasting glucose (FBG), total cholesterol, triglycerides (TG), high-density lipoproteins cholesterol (HDL-C), low-density lipoproteins cholesterol (LDL-C), and free fatty acids were determined using an automatic biochemical analyzer in the presence of quality control samples.

### 2.4. Dietary Assessment

Food frequency questionnaires with 10 food items were used to assess participants’ average consumption frequency of foods and beverages during the previous year based on personal diet and living habits, including milk (primarily liquid cow’s milk), salt, fruits, vegetables, red meat, seafood, eggs, soy products, nuts, and sugar-sweetened beverages [17,18,19,20]. Milk intake was categorized into five frequency ranges: 0 times per month, ≤1 time per month (250 mL per serving), >1 time per week, 2–4 times per week, 1 time per day, and ≥2 times per day.

### 2.5. Sleep Quality

The PSQI measures sleep quality and is used to detect sleep disorders in research and clinical settings. Its validity and reliability have been ascertained previously. PSQI includes seven domains: sleep quality, sleep latency, sleep duration, sleep efficiency, sleep disturbances, use of sleep medications, and daytime dysfunction. The scale ranges from 0 to 3 on each domain, with higher scores indicating a more severe situation. PSQI global scores are the summation of seven domain scores, and a PSQI global score of ≥5 indicates the presence of a sleep disorder [21,22].

### 2.6. Statistical Analysis

Tertile analysis found that the median milk intake of participants coincided with the lowest tertile, and more than two-thirds of participants’ milk intake was concentrated in the once-per-month group (Appendix A). Thus, milk intake was divided into two groups based on the distribution and median of intake: Rare (≤62.5 mL per week) and Regular (>62.5 mL per week) [23]. Low sleep quality was defined as a sleep quality score ≥ 2, and a longer sleep latency score was defined as a sleep latency score ≥ 2. Low sleep efficiency was defined as sleep efficiency < 85%. Short sleep duration was defined as sleep duration < 7 h. Sleep disturbances were defined as the onset of subjective sleep disturbances within the previous month. Sleep medication was defined as the use of sleep medication in the previous month. Daytime dysfunction was defined as the onset of daytime dysfunction within the previous month.

Data with a normal distribution were expressed as mean ± standard deviation, while those with a skewed distribution were expressed as the median ± interquartile range. Categorical data were expressed as numbers and percentages. The Mann−Whitney rank sum and chi-square test were used to analyze differences in continuous and categorical variables.

Multivariate-adjusted logistic regression models were used to assess the associations between milk intake and sleep disorders, as well as their specific dimensions. Odds rations (ORs) and their 95% confidence intervals (Cis) were calculated by using the rare milk intake as the reference category. In secondary analyses, multivariate-adjusted linear regression models were conducted to examine the impact of milk intake on sleep disorders and their specific dimensions, as well as blood physiological factors (ALT, AST, LDH, FBG, TC, TG, HDL, LDL, FFA). In both of the multivariable analyses, the crude model included milk intake only as the independent variable. Model 1 was adjusted for age, gender, and BMI. Model 2 was further adjusted for lifestyle factors including smoking, alcohol consumption, and dietary confounders including salt, fruit intake, vegetable intake, seafood intake, red meat intake, egg intake, soy product intake, nut intake, and sugar-sweetened beverage intake. Model 3 was a full model with additional adjustments for clinical factors including hypertension, diabetes, coronary heart disease, anxiety, and depression.

Subgroup analyses were performed by using logistic regression model (Model 4) in the strata of age (<65 y, ≥65 y), gender, BMI (<24, ≥24), smoking (yes, no), alcohol consumption (yes, no), hypertension (yes, no), dyslipidemia (yes, no), coronary heart disease (yes, no), anxiety (yes, no), and depression (yes, no) to determine the findings’ consistency. All subgroup analyses included adjustments same to those in the logistic regression model, except for the stratifying factors. Interactions between intake of milk and subgroups were tested by adding interaction product terms in the logistic regression models. Data were analyzed using the R programming language (version 4.2.2), and the significance level was set to 0.05.

## 3. Results

### 3.1. Baseline Characteristics

Participants’ data on anthropometric and socio-demographic information, dietary intake, lifestyle, clinical disease, sleep behavior, medications, and biochemical indicators of the study population stratified by milk intake are displayed in Table 1. Participants with a regular milk intake had a higher intake of fruits (<0.001), eggs (<0.01), nuts (<0.01), sugar-sweetened beverages (<0.01), less sleep disturbance (<0.001), and lower triglycerides (0.021) compared to those with rare intake. The prevalence of sleep disturbances decreased significantly with increasing milk intake (Figure 2). Milk intake was not significantly associated with the prevalence of sleep disorders (Appendix A). The results of the correlation analysis showed that scores of PSQI and sleep disturbances were negatively correlated with milk intake (Appendix A).

### 3.2. Association between Milk Intake and Sleep Disorders

Table 2 and Appendix A display the ORs with 95% CIs of sleep disorders and their specific dimensions for the comparison between regular and rare milk intake groups. In the crude model, participants with a regular milk intake had lower odds of sleep disorders (OR: 0.72, 95%; CI: 0.52, 0.99) compared to those with rare intake (≤62.5 mL per week). However, the association was not significant in the fully adjusted model. For specific dimensions, participants with a regular milk intake (>62.5 mL per week) had a lower risk of sleep disturbances (OR: 0.49, 95%; CI: 0.28, 0.87) than those with rare intake (≤62.5 mL per week) in the fully adjusted model.

Associations between milk intake and scores of PSQI assessed by the multivariate linear regressions are displayed in Table 3. In the fully adjusted model, no significant associations were found for milk intake and total scores of PSQI. For specific dimensions, increased intake of milk was significantly associated with lower scores of PSQI for sleep quality (β: −0.045, 95%; CI: −0.083, −0.007) and sleep disturbances (β: −0.059, 95%; CI: −0.090, −0.029) in the fully adjusted model.

### 3.3. Association between Milk Intake and Biochemical Parameters

The association between milk intake and clinical biochemical parameters is shown in Table 4. There was a significant inverse association between milk intake and triglyceride levels (β: −0.011, 95%; CI: −0.037, −0.004), and this association remained statistically significant (β: −0.020, 95%; CI: −0.037, −0.003) in the full model.

### 3.4. Subgroup Analysis

The subgroup analyses revealed milk intake was inversely associated with sleep disorders and disturbances among men (OR: 0.59, 95%; CI: 0.39, 0.89), whereas no association was found among women. Inverse associations between milk intake and sleep disorders and disturbances were also observed in older adults (OR: 0.58, 95%; CI: 0.36, 0.94), whereas there was no association with those under 65. The subgroup analysis showed no significant interaction between milk intake and stratifying factors (Figure 3 and Figure 4).

## 4. Discussion

We aimed to determine the association of milk intake with sleep disorders and their specific indicators, and our significant finding was that regular milk intake was associated with a lower prevalence of sleep disorders and disturbances. Furthermore, we found that regular milk intake was more favorably linked with a lower prevalence of sleep disorders and disturbances in male and older populations. In addition, we found a significant negative correlation between milk intake and serum triglyceride levels.

Our primary finding in the present study was a positive link between the decreased prevalence of sleep disorders and disturbances and milk intake. Clinical research and laboratory-based studies show that milk improves sleep disorders [24,25]. A cross-sectional study found that PSQI scores improved significantly after yogurt ingestion [26]. Kitano et al. found that commercial milk intake facilitated falling asleep in older adults [10]. A clinical trial demonstrated significantly increased sleep efficiency after ingesting melatonin-rich milk [27]. In addition, an animal study found that nocturnal milk consumption produced sedative and anxiolytic-like effects in mice, enhancing pentobarbital-induced sleep comparable to the effect induced by the benzodiazepine diazepam [25]. There is a tight physiological connection between milk and sleep. Studies showed that melatonin, whey protein, and bioactive peptides in milk might promote sleep [12,25]. Several studies showed that milk contains melatonin, essential in regulating the sleep−wake cycle [28,29,30]. Melatonin works in the suprachiasmatic nucleus of the hypothalamus to attenuate wake-up boosting signals in the circadian clock, thereby promoting sleep and successfully treating sleep disorders [28,31]. Moreno-Fernandez et al. found that the administration of goat milk improved melatonin levels in rats [32]. Cow’s milk-derived whey protein is an excellent source of tryptophan. This amino acid is the indirect precursor of serotonin used for melatonin production [33]. In addition, milk is rich in biologically active peptides (including peptides with opioid and opioid antagonist activities) that may be related to the opioid system or GABAergic nerves that regulate sleep. For example, α−s1−casein hydrolysate exhibits benzodiazepine activity on GABA(A) receptors, which increases the duration of pentobarbital-induced sleep in mice [12,13]. These findings support our findings that participants with regular milk intake had lower rates of sleep disorders and disturbances.

However, after adjusting for confounding factors, milk intake was not associated with sleep disorders but was significantly associated with sleep disturbances. Studies showed that milk contains low concentrations of bioactive compounds (such as melatonin, with a content of 0.004–0.056 ng/mL), which may be insufficient to improve sleep disorders [29,34]. In addition, a recent cross-sectional study found that the sleep-promoting effect of milk may significantly reduce sleep disturbances, such as waking up easily, early awakening, feeling hot or cold, and experiencing nightmares [35]. Melatonin in milk has a neuroprotective effect and can reduce inflammation [36]. A prospective cohort study found that early waking in women is associated with increased inflammatory factors [37]. Thus, milk may reduce sleep disturbances through its neuroprotective effect. In addition, we found a significant linear trend between milk intake and subjective sleep quality scores. A cross-sectional study found that the higher the frequency of milk consumption in Japanese athletes, the lower the risk of subjective sleep quality [38]. There is scant literature on the association between sleep disturbances, sleep quality, and milk intake. Therefore, our findings contribute to this developing area of research [10]. Furthermore, our study found that individuals with a milk intake of 62.5 mL or more per week had better sleep quality compared to those with a milk intake less than 62.5 mL. Similar findings have been reported in previous studies. For instance, Min et al. found that individuals with a milk intake frequency of 1–2 times per week had better sleep quality compared to individuals with no milk intake per week [39]. Mozaffarian et al. found that as milk intake frequency (never, rarely, weekly, daily) increased, the probability of individuals having better sleep quality also increased [40]. Meanwhile, in a study of food consumption by Chinese residents, it was found that the average intake of milk in normal rural areas in China was 9.1 g/d. The frequency of milk intake decreased with aging [41]. The population of this study was basically middle-aged and elderly populations (Over 90% are older than 50 years) [42]. Therefore, the milk intake of 62.5 mL/week in this study was in line with the consumption status of normal rural population.

Secondary studies showed that regular milk intake was associated with a lower prevalence of sleep disorders and disturbances in only older and male populations. There is substantial evidence supporting our findings. Studies showed that older adults more often experience less sleep quantity and quality than younger adults, and diet may influence sleep outcomes in older adults [43]. Cross-sectional studies and controlled clinical trials have generally found that dairy products improve sleep quality in older adults [10,44]. A longitudinal study found that higher milk intake was associated with reduced sleep initiation problems in older men [23]. In addition, one clinical study suggested that milk consumption may be associated with a longer night sleep in men [11]. However, the mechanisms behind this gender difference are worth discussing. First, women are subject to more psychological and social pressures than men, which can result in more frequent awakenings during the night [45,46]. Second, women experience physiological changes in middle to old age, including rapid declines in estrogen levels during menopause, which can reduce the availability of serotonin in the brain [38,45,47,48]. Therefore, our results should be interpreted with caution. Studies showed that endocrine changes characteristic of perimenopause could begin around age 45 [49], and our study included 98.5% of women aged 45 and over. If the age distribution of the participants had been balanced, the results might have differed.

Finally, our study found that the serum triglyceride levels were negatively and linearly correlated with milk intake. The association between triglycerides and milk is a matter of debate. However, substantial evidence suggests that the whey protein in milk decreases serum triglyceride levels [50,51,52]. Yogurt and fermented milk reduced the triglyceride content in rats with spontaneous hypertension and might be an adjuvant to reducing hypertension and hyperlipidemia [53,54]. Santesso et al. found that participants with higher protein intake had lower serum triglyceride levels than those with lower protein intake [55]. These results agree with our findings. Studies showed that triglycerides were associated with cardiovascular disease and sleep disorders [56]. In a longitudinal analysis, Peila et al. found that changes from restful sleep to insomnia were associated with an increased incidence of metabolic syndrome and high triglyceride levels (≥150 mg/dL) [57]. Shigiyama et al. demonstrated that the liver triglyceride content of mice in a chronic sleep deprivation model was significantly increased [58]. Interestingly, excessive sleep duration also increases the incidence of metabolic syndrome and the incidence of high triglyceride levels [59]. A U-shaped relationship exists between sleep duration, metabolic syndrome, and triglyceride [60]. Nevertheless, the relationship between milk, triglyceride, and sleep requires further study.

This study has several strengths. First, we evaluated the effects of milk on sleep disorders and their dimensions. Second, we adjusted for known and potential risk factors. Third, we examined the associations using basic information stratification. There are also some limitations. First, a cross-sectional study cannot draw causal inferences. Furthermore, due to the use of a Food Frequency Questionnaire (FFQ) for dietary assessment and the Pittsburgh Sleep Quality Index (PSQI) Scale for sleep quality evaluation, measurement errors in milk intake and sleep quality were inevitable. However, these questionnaires have been validated in prior studies to be effective [19,20,61]. Moreover, accounting for cultural and socioeconomic differences may affect the overall interpretation of the results, and our results may only be generalizable to populations residing in the eastern coastal regions of China. Cross-national and multiethnic studies to further explore the association are warranted. In addition, although significant associations were observed, given the low milk consumption per week among 66% of our sample population, further research is warranted to confirm this finding. Finally, although we adjusted for various confounding variables, we cannot rule out the possibility that the observed associations are due to unmeasured confounders (such as neuropsychiatric disorders).

In conclusion, our findings suggest that regular milk intake is associated with a lower prevalence of sleep disorders and disturbances. Milk intake was significantly associated with sleep disorders and disturbances, especially in older adults and males. These findings suggest that cow’s milk may be an effective natural sleep aid for managing sleep-related disorders [25]. In addition, milk intake and serum TG levels had a significant negative correlation. TG is a risk factor for sleep disorders and may act as a physiological link between milk and sleep disorders [58,62]. However, prospective and biological research is needed to investigate the physiological link between milk intake and the risk of sleep disorders.

## Figures and Tables

**Figure 1 nutrients-15-04079-f001:**
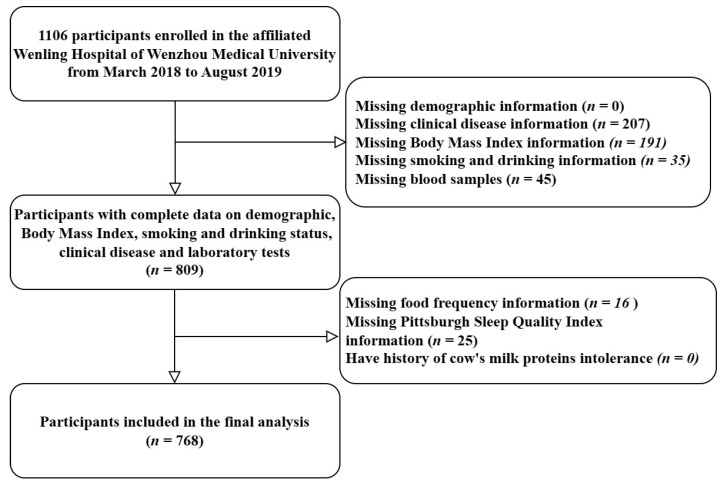
Flowchart of study participant selection.

**Figure 2 nutrients-15-04079-f002:**
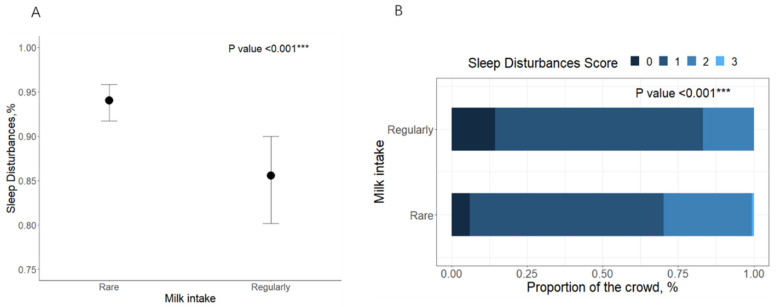
The prevalence of sleep disturbances in milk intake groups. Note: (**A**) the percentage of sleep disturbances and 95% CI in milk intake groups. (**B**) The distribution of sleep disturbance scores across milk intake groups. *** *p* < 0.001.

**Figure 3 nutrients-15-04079-f003:**
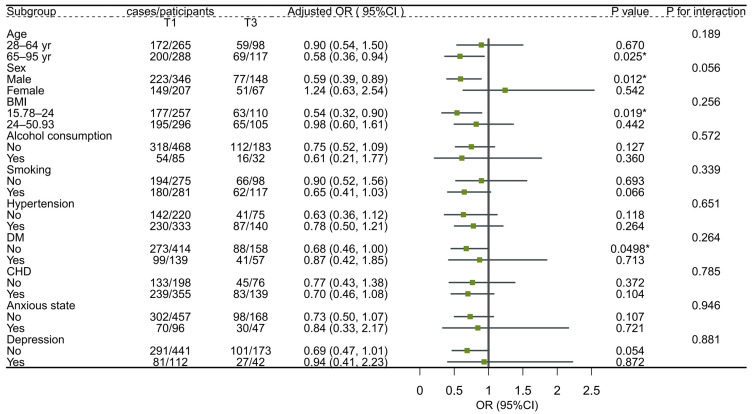
Subgroup analyses of the associations (ORs, 95% CIs) between milk intake and sleep disorders among 768 participants with regular vs. rare milk intake. Note: logistic regression models adjusted for age, gender, BMI, smoking, alcohol consumption, dietary confounders (salt, fruit intake, vegetable intake, seafood intake, red meat intake, egg intake, soy product intake, nuts intake, sugar-sweetened beverages intake) and clinical factors (hypertension, diabetes, coronary heart disease, anxiety, and depression). In each case, the model is not adjusted for the stratification variable. * *p* < 0.05.

**Figure 4 nutrients-15-04079-f004:**
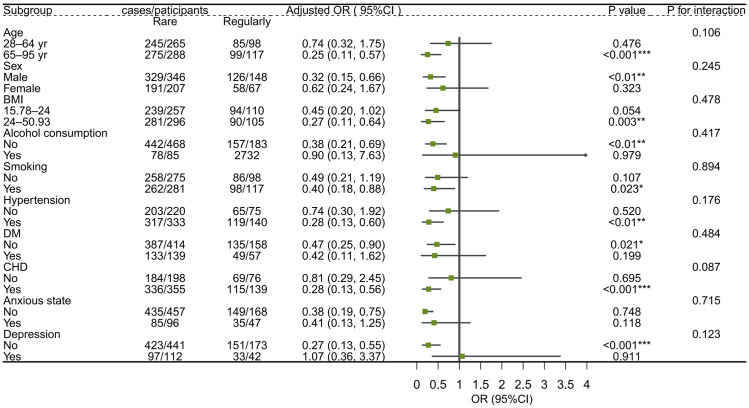
Subgroup analyses of the associations (ORs, 95% CIs) between milk intake and sleep disturbance among 768 participants with regular vs. rare milk intake. Note: logistic regression models adjusted for age, gender, BMI, smoking, alcohol consumption, dietary confounders (salt, fruit intake, vegetable intake, seafood intake, red meat intake, egg intake, soy product intake, nuts intake, sugar-sweetened beverages intake) and clinical factors (hypertension, diabetes, coronary heart disease, anxiety, and depression). In each case, the model is not adjusted for the stratification variable. * *p* < 0.05, ** *p* < 0.01, *** *p* < 0.001.

**Table 1 nutrients-15-04079-t001:** Baseline characteristics of study participants compared between the regular and rare milk intake groups.

Value	Milk Intake		*p*
	Rare (≤62.5 mL Per Week), *n* = 553	Regularly (>62.5 mL Per Week), *n* = 215	
**Socio demographics**			
Male, %	346 (62.23)	148 (68.84)	0.123
Age (≥65 years), %	288 (52.08)	117 (54.42)	0.615
BMI (≥24 kg/m^2^), %	296 (53.53)	105 (48.84)	0.277
**Lifestyle risk factors**			
alcohol consumption, %	85 (15.29)	32 (14.88)	0.955
smoking, %	279 (50.18)	117 (54.42)	0.364
Salt intake (<6 g/d), %	155 (28.17)	63 (29.30)	0.793
Fruit intake, %	324 (58.27)	178 (82.79)	<0.001 ***
Vegetable intake, %	548 (98.56)	214 (99.53)	0.870
Red meat intake, %	531 (95.50)	208 (96.74)	0.794
Seafood intake, %	498 (89.57)	200 (93.02)	0.253
Egg intake, %	446 (80.22)	193 (89.77)	<0.01 **
Soy product intake, %	370 (66.55)	150 (69.77)	0.500
Nuts intake, %	149 (26.80)	82 (38.14)	<0.01 **
Sugar-sweetened beverages intake, %	83 (14.93)	53 (24.65)	<0.01 **
**Clinical assessment**			
Hypertension, %	333 (60.22)	140 (65.12)	0.242
Diabetes mellitus, %	139 (25.14)	56 (26.05)	0.867
CHD, %	355 (64.20)	139 (64.65)	0.973
Anxious state, %	96 (17.36)	47 (21.86)	0.182
Depression, %	112 (20.25)	42 (19.53)	0.806
**PSQI Global Scores**	6 (4, 9)	6 (3, 8)	0.068
**Sleep Disorders, %**	372 (67.27)	128 (59.53)	0.053
Low Sleep Quality, %	138 (24.95)	45 (20.93)	0.280
Low Sleep Efficiency, %	278 (50.27)	112 (52.09)	0.709
Long Sleep Latency, %	141 (25.50)	44 (20.47)	0.218
Short Sleep Duration, %	333 (60.22)	140 (65.12)	0.242
Sleep Disturbances, %	520 (94.03)	184 (85.58)	<0.001 ***
Daytime Dysfunction, %	395 (71.43)	141 (65.58)	0.134
Used Sleep Medication, %	20 (3.62)	9 (4.19)	0.872
**Medication factors**			
Anti-hypertensive drugs, %	371 (67.09)	146 (67.91)	0.896
Hypoglycemic drugs, %	129 (23.33)	49 (22.79)	0.950
**Biochemical Indicators**			
ALT, U/L	18.87 (13.80, 27.50)	20.00 (14.60, 28.60)	0.284
AST, U/L	21.40 (17.70, 27.50)	22.80 (18.50, 29.75)	0.127
LDH, U/L	190.00 (165.00, 220.00)	187.00 (164.00, 217.00)	0.693
FBG, mmol/L	5.19 (4.73, 6.15)	5.14 (4.63, 6.19)	0.497
TC, mmol/L	4.35 (3.58, 5.13)	4.24 (3.53, 5.19)	0.757
TG, mmol/L	1.44 (1.11, 2.08)	1.34 (1.00, 1.85)	0.021 *
HDL-C, mmol/L	1.00 (0.85, 1.18)	1.04 (0.88, 1.22)	0.139
LDL-C, mmol/L	2.96 (2.32, 3.52)	2.93 (2.33, 3.50)	0.879
FFA, µmol/L	364.00 (247.00, 514.00)	361.00 (244.50, 542.00)	0.863

Note: Variables using percentages were reported as the chi-square test between Rare and Regularly. Other data were reported as Mann−Whitney rank sum test between Rare and Regularly. * *p* < 0.05, ** *p* < 0.01, *** *p* < 0.001.

**Table 2 nutrients-15-04079-t002:** Odds ratios of sleep disorders or PSQI components and corresponding 95% confidence intervals for the comparison between regular and rare milk intake groups.

	Milk Intake	Cases/ Participants	Crude Model		Model 1		Model 2		Model 3	
			OR (95% CI)	*p* Value	OR (95% CI)	*p* Value	OR (95% CI)	*p* Value	OR (95% CI)	*p* Value
**Sleep Disorder**	Rare	372/553	Reference		Reference		Reference		Reference	
	Regularly	128/215	0.72 (0.52, 0.99)	0.045 *	0.73 (0.53, 1.02)	0.064	0.74 (0.53, 1.05)	0.092	0.72 (0.51, 1.03)	0.068
**PSQI Component**										
**Low Sleep Quality**	Rare	138/553	Reference		Reference		Reference		Reference	
	Regularly	45/215	0.80 (0.54, 1.16)	0.240	0.82 (0.55, 1.20)	0.307	0.85 (0.56, 1.27)	0.427	0.84 (0.55, 1.28)	0.422
**Low Sleep Efficiency**	Rare	278/553	Reference		Reference		Reference		Reference	
	Regularly	112/215	1.08 (0.79, 1.48)	0.650	1.09 (0.79, 1.51)	0.605	0.97 (0.69, 1.38)	0.881	0.97 (0.69, 1.38)	0.871
**Long Sleep Latency**	Rare	141/553	Reference		Reference		Reference		Reference	
	Regularly	44/215	0.92 (0.67, 1.26)	0.608	0.95 (0.69, 1.31)	0.746	0.85 (0.60, 1.19)	0.346	0.85 (0.60, 1.19)	0.334
**Short Sleep Duration**	Rare	333/553	Reference		Reference		Reference		Reference	
	Regularly	140/215	1.23 (0.89, 1.72)	0.211	1.27 (0.92, 1.78)	0.153	1.24 (0.88, 1.77)	0.218	1.22 (0.86, 1.74)	0.259
**Sleep Disturbances**	Rare	520/553	Reference		Reference		Reference		Reference	
	Regularly	184/215	0.38 (0.22, 0.63)	<0.001 ***	0.38 (0.22, 0.64)	<0.001 ***	0.50 (0.28, 0.87)	0.014 *	0.49 (0.28, 0.87)	0.015 *
**Daytime Dysfunction**	Rare	395/553	Reference		Reference		Reference		Reference	
	Regularly	141/215	0.76 (0.55, 1.07)	0.114	0.79 (0.56, 1.11)	0.163	0.84 (0.59, 1.21)	0.349	0.83 (0.58, 1.19)	0.316
**Used Sleep Medication**	Rare	20/553	Reference		Reference		Reference		Reference	
	Regularly	9/215	1.16 (0.50, 2.53)	0.710	1.17 (0.49, 2.58)	0.704	1.20 (0.48, 2.81)	0.677	1.08 (0.42, 2.60)	0.876

Note: Model 1: adjusted for age, gender, and BMI. Model 2: Model 1 plus an adjustment for lifestyle factors, including smoking, alcohol consumption, and dietary confounders (salt, fruit intake, vegetable intake, seafood intake, red meat intake, egg intake, soy product intake, nuts intake, and sugar-sweetened beverages intake). Model 3: Model 2 plus an adjustment for clinical factors (hypertension, diabetes, coronary heart disease, anxiety, and depression). * *p* < 0.05, *** *p* < 0.001.

**Table 3 nutrients-15-04079-t003:** Coefficients of PSQI global scores or scores of PSQI components and corresponding 95% confidence intervals for the comparison between regular and rare milk intake groups.

	Milk Intake	Crude Model		Model 1		Model 2		Model 3	
		β (95% CI)	*p* Value	β (95% CI)	*p* Value	β (95% CI)	*p* Value	β (95% CI)v	*p* Value
**PSQI Global Score**	Rare, *n* = 553								
	Regularly, *n* = 215	−0.029 (−0.061, 0.002)	0.069	−0.025 (−0.056, 0.006)	0.109	−0.021 (−0.053, 0.011)	0.198	−0.023 (−0.055, 0.009)	0.155
**Scores of PSQI Components**									
**Low Sleep Quality**	Rare, *n* = 553								
	Regularly, *n* = 215	−0.056 (−0.093, −0.018)	<0.01 **	−0.053 (−0.090, −0.016)	<0.01 **	−0.044 (−0.082, −0.006)	0.024 *	−0.045 (−0.083, −0.007)	0.020 *
**Low Sleep Efficiency**	Rare, *n* = 553								
	Regularly, *n* = 215	0.007 (−0.059, 0.074)	0.830	0.009 (−0.056, 0.075)	0.780	−0.006 (−0.073, 0.062)	0.864	−0.007 (−0.074, 0.060)	0.838
**Long Sleep Latency**	Rare, *n* = 553								
	Regularly, *n* = 215	−0.028 (−0.083, 0.026)	0.304	−0.029 (−0.083, 0.024)	0.286	−0.031 (−0.087, 0.025)	0.277	−0.031 (−0.087, 0.024)	0.269
**Short Sleep Duration**	Rare, *n* = 553								
	Regularly, *n* = 215	0.002 (−0.058, 0.055)	0.950	0.003 (−0.054, 0.059)	0.927	0.010 (−0.049, 0.068)	0.749	0.007 (−0.052, 0.066)	0.811
**Sleep Disturbances**	Rare, *n* = 553								
	Regularly, *n* = 215	−0.074 (−0.103, −0.044)	<0.001 ***	−0.071 (−0.101, −0.042)	<0.001 ***	−0.057 (−0.087, −0.026)	<0.001 ***	−0.059 (−0.090, −0.029)	<0.001 ***
**Daytime Dysfunction**	Rare, *n* = 553								
	Regularly, *n* = 215	−0.030 (−0.008, 0.020)	0.236	−0.023 (−0.073, 0.026)	0.358	−0.007 (−0.059, 0.044)	0.784	−0.011 (−0.062, 0.040)	0.677
**Used Sleep Medication**	Rare, *n* = 553								
	Regularly, *n* = 215	−0.004 (−0.025, 0.017)	0.706	−0.004 (−0.024, 0.017)	0.721	−0.005 (−0.027, 0.016)	0.622	−0.008 (−0.029, 0.013)	0.466

Note: Model 1: adjusted for age, gender, and BMI. Model 2: Model 1 plus an adjustment for lifestyle factors, including smoking, alcohol consumption, and dietary confounders (salt, fruit intake, vegetable intake, seafood intake, red meat intake, egg intake, soy product intake, nuts intake, and sugar-sweetened beverages intake). Model 3: Model 2 plus an adjustment for clinical factors (hypertension, diabetes, coronary heart disease, anxiety, and depression). * *p* < 0.05, ** *p* < 0.01, *** *p* < 0.001.

**Table 4 nutrients-15-04079-t004:** Linear regression of index related to blood physiological factors associated with milk intake.

	Milk Intake	Crude Model		Model 1		Model 2		Model 3	
		β (95% CI)	*p* Value	β (95% CI)	*p* Value	β (95% CI)	*p* Value	β (95% CI)	*p* Value
**ALT, U/L**	Rare, *n* = 553								
	Regularly, *n* = 215	−0.002 (−0.014, 0.011)	0.764	−0.002 (−0.015, 0.010)	0.734	−0.004 (−0.017, 0.009)	0.547	−0.003 (−0.016, 0.010)	0.655
**AST, U/L**	Rare, *n* = 553								
	Regularly, *n* = 215	−0.004 (−0.018, 0.010)	0.585	−0.005 (−0.020, 0.009)	0.475	−0.005 (−0.020, 0.010)	0.544	−0.003 (−0.018, 0.012)	0.699
**LDH, U/L**	Rare, *n* = 553								
	Regularly, *n* = 215	−0.005 (−0.019, 0.010)	0.515	−0.006 (−0.021, 0.007)	0.429	−0.003 (−0.018, 0.012)	0.710	−0.001 (−0.016, 0.013)	0.850
**FBG, mmol/L**	Rare, *n* = 553								
	Regularly, *n* = 215	0.004 (−0.016, 0.023)	0.718	0.006 (−0.013, 0.026)	0.531	0.012 (−0.008, 0.033)	0.233	0.006 (−0.011, 0.023)	0.502
**TC, mmol/L**	Rare, *n* = 553								
	Regularly, *n* = 215	0.004 (−0.001, 0.010)	0.134	0.004 (−0.001, 0.010)	0.134	0.004 (−0.002, 0.010)	0.159	0.005 (−0.001, 0.011)	0.128
**TG, mmol/L**	Rare, *n* = 553								
	Regularly, *n* = 215	−0.011 (−0.037, 0.004)	0.015 *	−0.017 (−0.033, −0.004)	0.045 *	−0.019 (−0.036, 0.002)	0.030 *	−0.020 (−0.037, −0.003)	0.023 *
**HDL, mmol/L**	Rare, *n* = 553								
	Regularly, *n* = 215	0.016 (−0.006, 0.037)	0.147	0.016 (−0.005, 0.037)	0.132	0.020 (−0.002, 0.041)	0.072	0.021 (−0.001, 0.042)	0.059
**LDL, mmol/L**	Rare, *n* = 553								
	Regularly, *n* = 215	0.004 (−0.021, 0.030)	0.727	0.010 (−0.015, 0.034)	0.439	0.005 (−0.020, 0.031)	0.685	0.007 (−0.018, 0.033)	0.580
**FFA, µmol/L**	Rare, *n* = 553								
	Regularly, *n* = 215	0.004 (−0.013, 0.020)	0.644	0.004 (−0.012, 0.021)	0.604	0.001 (−0.016, 0.018)	0.923	−0.002 (−0.018, 0.015)	0.853

Note: Model 1: Adjusted for age, gender, and BMI. Model 2: Model 1 plus an adjustment for lifestyle factors, including smoking, alcohol consumption, and dietary confounders (salt, fruit intake, vegetable intake, seafood intake, red meat intake, egg intake, soy product intake, nuts intake, and sugar-sweetened beverages intake). Model 3: Model 2 plus an adjustment for clinical factors (hypertension, diabetes, coronary heart disease, anxiety, and depression). * *p* < 0.05.

## Data Availability

The data that support the findings of this study are available from the corresponding author upon reasonable request.

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
