# Peer review of "Associations between Milk Intake and Sleep Disorders in Chinese Adults: A Cross-Sectional Study"

_nutrients, 2023, doi:10.3390/nu15184079_

Round 1
Reviewer 1 Report (New Reviewer)
The article "Associations between milk intake and sleep disorders in Chinese adults: a cross-sectional study" attempts to examine the relationship between milk intake and sleep disorders in the Chinese adult population. Although the sample size is impressive and the approach to analysis is comprehensive, the study has several important limitations that should be addressed in a revised version.
The most serious limitation is the cross-sectional nature of the study, which does not allow causality to be established, only correlation. The authors should explicitly note this limitation in the revised version and possibly include additional studies that may allow a better understanding of causal relationships.
In addition, the lack of placebo control and possible self-reporting errors pose a serious threat to the reliability of the results. The authors must include these aspects in the revised version to ensure that their conclusions are solidly grounded in the data.
Also, failure to account for cultural and socioeconomic differences may affect the overall interpretation of the results, given the complexity of dietary effects on sleep quality in different social and economic settings.
Overall, the article provides valuable insights, but needs serious revision to truly contribute to the literature on the effects of milk consumption on sleep disorders. I would expect that the authors would undertake to address the demonstrated limitations in a revised version to provide readers with more precise and reliable conclusions.
Author Response
We sincerely thank you for your insightful and constructive comments, and we have improved the quality of our article according to the valuable suggestions. We would like to submit the revised version of our manuscript entitled “Associations between milk intake and sleep disorders in Chinese adults: a cross-sectional study” (ID: nutrients-2567010). The revisions in the manuscript are highlighted with yellow, and we hope our revisions will meet nutrients’ approval. Our point-by-point responses to the comments are listed below:
- The most serious limitation is the cross-sectional nature of the study, which does not allow causality to be established, only correlation. The authors should explicitly note this limitation in the revised version and possibly include additional studies that may allow a better understanding of causal relationships.
Response: Thank you for noting this issue. We have revised the manuscript accordingly to accentuate this limitation. (page 15, line 353)
First, a cross−sectional study cannot draw causal inferences. (page 15, line 353)
- In addition, the lack of placebo control and possible self-reporting errors pose a serious threat to the reliability of the results. The authors must include these aspects in the revised version to ensure that their conclusions are solidly grounded in the data.
Response: We are very thankful for comments 1 and 2; both of which addressed important issues about the reliability of the results. Indeed, the current study lacked a placebo control, as it is a cross-sectional study rather than a randomized controlled trial. Furthermore, the possible self-report errors that exists is a limitation that should be mentioned to ensure readers are well aware of the reliability of the results. (page 15, line 354-357)
Furthermore, due to the use of a Food Frequency Questionnaire (FFQ) for dietary assessment and the Pittsburgh Sleep Quality Index (PSQI) Scale for sleep quality evaluation, measurement errors in milk intake and sleep quality were inevitable. However, these questionnaires have been validated in prior studies to be effective[1-3]. (page 15, line 354-357)
- Also, failure to account for cultural and socioeconomic differences may affect the overall interpretation of the results, given the complexity of dietary effects on sleep quality in different social and economic settings.
Response: Thank you for noting this important issue as well! In the present study, several limitations regarding the population sample examined should be noted. The current sample consists entirely of Chinese patients. Though the findings reported are similar to those reported in western cultures (Knowlden, A.P et al., 2016)[4]. Cultural differences, as well as socioeconomic differences should be accounted for when interpreting the findings in the present study (for details see Hepsomali P et al., 2021)[5]. This content has been added in Discussion in the revised manuscript: (page 15, line 358-361)
Also, accounting for cultural and socioeconomic differences may affect the overall interpretation of the results, our results may only be generalizable to population residing in the eastern coastal regions of China. Cross-national and multiethnic studies to further explore the association are warranted. (page 15, line 358-361)
- Luo, T.Y.; X.H. Liu; T.Y. Dai; X.M. Liu; Q. Zhang; J.Z. Dong. Ideal Cardiovascular Health Metrics and Coronary Artery Calcification in Northern Chinese Population: A Cross-sectional Study. Biomed Environ Sci. 2016. 29, 475-83.
- Zhuang, M.; Z. Yuan; L. Lin; B. Hu; X. Wang; Y. Yang; X. Chen; L. Jin; M. Lu; W. Ye. Reproducibility and relative validity of a food frequency questionnaire developed for adults in Taizhou, China. PLoS One. 2012. 7, e48341.
- Spira, A.P.; S.A. Beaudreau; K.L. Stone; E.J. Kezirian; L.Y. Lui; S. Redline; S. Ancoli-Israel; K. Ensrud; A. Stewart. Reliability and validity of the Pittsburgh Sleep Quality Index and the Epworth Sleepiness Scale in older men. J Gerontol A Biol Sci Med Sci. 2012. 67, 433-9.
- Knowlden, A.P.; M. Burns; A. Harcrow; M.E. Shewmake. Cross-sectional analysis of food choice frequency, sleep confounding beverages, and psychological distress predictors of sleep quality. Int J Adolesc Med Health. 2016. 30.
- Hepsomali, P.; J.A. Groeger. Diet, Sleep, and Mental Health: Insights from the UK Biobank Study. Nutrients. 2021. 13.

Reviewer 2 Report (Previous Reviewer 3)
The authors have addressed my concerns and the manuscript is acceptable for publication.
Author Response
We sincerely thank you for your insightful and constructive comments, and we have improved the quality of our article according to the valuable suggestions. We would like to submit the revised version of our manuscript entitled “Associations between milk intake and sleep disorders in Chinese adults: a cross-sectional study” (ID: nutrients-2567010). The revisions in the manuscript are highlighted with yellow, and we hope our revisions will meet nutrients’ approval. Our point-by-point responses to the comments are listed below:
The authors have addressed my concerns and the manuscript is acceptable for publication.
Response: Many thanks for your positive feedback.

Round 2
Reviewer 1 Report (New Reviewer)
Current version can be accepted.
Author Response
We sincerely thank you for your insightful and constructive comments, and we have improved the quality of our article according to the valuable suggestions. We would like to submit the revised version of our manuscript entitled “Associations between milk intake and sleep disorders in Chinese adults: a cross-sectional study” (ID: nutrients-2567010). The revisions in the manuscript are highlighted with yellow, and we hope our revisions will meet nutrients’ approval. Our point-by-point responses to the comments are listed below:
Current version can be accepted.
Response: Many thanks for your positive feedback.

This manuscript is a resubmission of an earlier submission. The following is a list of the peer review reports and author responses from that submission.
Round 1
Reviewer 1 Report
The work is interesting and the topic is stimulating. However, in my opinion, in the paper it should be indicated that it is a study on cow's milk and not on goat's milk , for example.
Subjects with history of cow's milk proteins intolerance should be excluded, because they can have sleep disturbances.
In my opinion, only a minor editing of English language is required
Author Response
We sincerely thank you for your insightful and constructive comments, and we have improved the quality of our article according to the valuable suggestions. We would like to submit the revised version of our manuscript entitled “Associations between milk intake and sleep disorders in Chinese adults: a cross-sectional study” (ID: nutrients-2381985). The revisions in the manuscript are highlighted with yellow, and we hope our revisions will meet nutrients’ approval. Our point-by-point responses to the comments are listed below:
- However, in my opinion, in the paper it should be indicated that it is a study on cow's milk and not on goat's milk, for example.
Response:
Thank you for your comment. The milk in the present study is liquid cow's milk. This content has been added in Materials and Methods in the revised manuscript: (page 4, line 125-129)
Food frequency questionnaires with 10 food items were used to assess participants' average consumption frequency of foods and beverages during the previous year based on personal diet and living habits, including milk (primarily liquid cow’s milk), salt, fruits, vegetables, red meat, seafood, eggs, soy products, nuts, and sugar−sweetened beverages [17-20]. (page 4, line 125-129)
- Subjects with history of cow's milk proteins intolerance should be excluded, because they can have sleep disturbances.
Response:
Thank you for your suggestion. As you suggested, we have excluded the subjects with the intolerance history of cow's milk protein in Materials and Methods in the revised manuscript: (page 2-3, line 95-100)
We excluded 338 subjects for the following reasons: i) lack of demographic and clinical disease information (n = 241); ii) lack of food frequency questionnaire or PSQI data (n = 220); iii) lack of blood samples (n = 45); iiii) have history of cow's milk proteins intolerance (n = 0), the final study sample comprised 768 participants (mean age 64.71 ± 10.51 years, male 64.32%). For detailed information on participant recruitment, please refer to Figure 1. (page 2-3, line 95-100)

Reviewer 2 Report
Thank you very much for allowing me to review the article entitled "Associations between milk intake and sleep disorders in Chinese adults: a cross-sectional study" (nutrients-2381985), submitted for possible publication in the "Nutrition and Public Health" section of the Special Issue "Roles of Dairy Intake in Health Development".
The topic of this work is of great relevance in public health, as sleep is related to the balance of the body and its alteration affects the individual's health status significantly. Additionally, an increase in sleep disorders has been reported in recent years.
There are two aims: to determine the association between milk and sleep disorders and to identify the specific indicators of sleep disorders that milk affects.
Comments:
In the introduction, the topic is presented clearly. However, while other studies on this topic are mentioned, it should be explained what this study could contribute to the previous knowledge, i.e. what hypothesis is being proposed.
In the methodology, it should be indicated whether a sample size calculation has been performed for this study. It should also be clarified how the participants were recruited for the study (whether they are people who come to the hospital for sleep problems or if it is a general population), as this could influence the results.
Results: The results are also presented, but the odds ratios (ORs) should be expressed with two decimal places since three decimal places are not common. In the tables, the reference level should be indicated as OR (Reference).
Please confirm whether Figures 2 and 3 evaluate sleep disorders independently of milk consumption, and whether milk consumption and adjustment variables are also analyzed.
Author Response
We sincerely thank you for your insightful and constructive comments, and we have improved the quality of our article according to the valuable suggestions. We would like to submit the revised version of our manuscript entitled “Associations between milk intake and sleep disorders in Chinese adults: a cross-sectional study” (ID: nutrients-2381985). The revisions in the manuscript are highlighted with yellow, and we hope our revisions will meet nutrients’ approval. Our point-by-point responses to the comments are listed below:
- In the introduction, the topic is presented clearly. However, while other studies on this topic are mentioned, it should be explained what this study could contribute to the previous knowledge, i.e. what hypothesis is being proposed.
Response:
Thank you for your suggestion. In Introduction, we have proposed the hypothesis as you suggested. (page 2, line 68-73)
Previous studies have examined the potential benefits of milk intake on sleep quality [14-16]. However, the association between specific indicators of sleep disorders and milk intake in Chinese adults are not fully understood. Therefore, we hypothesized that milk would improve sleep disorders, as well as specific indicators of sleep disorders. The present study has two aims: 1) to determine the association between milk and sleep disorders and 2) to identify the specific indicators of sleep disorders that milk affects. (page 2, line 68-73)
- In the methodology, it should be indicated whether a sample size calculation has been performed for this study. It should also be clarified how the participants were recruited for the study (whether they are people who come to the hospital for sleep problems or if it is a general population), as this could influence the results.
Response:
Thank you for your suggestion. We have calculated the theoretical sample size. The theoretical sample size of this study was set at 1,056 individuals to provide a specific relative precision of 2% (Type I error, 0.05; Type II error, 0.10), taking into account an anticipated 80% participation rate. In addition, the original plan for this study was to recruit a total of 3,000 in two phases. The first phase involved posting posters at Wenling Hospital to recruit 1500 people, and ultimately recruited 1106 people. Meanwhile, the study population are general patients from Wenling Hospital affiliated to Wenzhou Medical University. The revisions were added in Materials and Methods in the revised manuscript, the details for participants’ recruitment can be found in Figure1. (page 2-3, line 85-100)
The study participants were enrolled at baseline within an ongoing cohort in Zhejiang Province, China. An original plan for the cohort was to recruit 3,000 general patients at baseline. The first phase plan to recruit 1,500 persons by posting posters at Wenling Hospital affiliated to Wenzhou Medical University, Zhejiang Province between March 2018 and August 2019. Ultimately, a total of 1,106 general patients aged 25–95 years residing in Taizhou Wenling City were enrolled in this study, with a response rate of 73.73%. The theoretical sample size of this study was set at 1,056 individuals to provide a specific relative precision of 2% (Type I error, 0.05; Type II error, 0.10), taking into account an anticipated 80% participation rate. Epidemiology survey data included blood-based biochemical parameters, anthropometric measurements, socio−demographics, dietary intake, and lifestyle characteristics. We excluded 338 subjects for the following reasons: i) lack of demographic and clinical disease information (n = 241); ii) lack of food frequency questionnaire or PSQI data (n = 220); iii) lack of blood samples (n = 45); iiii) have history of cow's milk proteins intolerance (n = 0), the final study sample comprised 768 participants (mean age 64.71 ± 10.51 years, male 64.32%). For detailed information on participant recruitment, please refer to Figure 1. (page 2-3, line 85-100)
- Results: The results are also presented, but the odds ratios (ORs) should be expressed with two decimal places since three decimal places are not common. In the tables, the reference level should be indicated as OR (Reference).
Response:
Thank you for your comment. We have made corresponding revisions as you stated. (page 7, line 207-215; page 10, line 251-255; page 11, line 257-260)
Please confirm whether Figures 2 and 3 evaluate sleep disorders independently of milk consumption, and whether milk consumption and adjustment variables are also analyzed.
Response:
Thank you for presenting this point. In fact, the subgroup analysis for the prevalence of sleep disorders associated with the regular vs. rare intake of milk is presented in Figure 2 and Figure 3, respectively. The potential confounding variables has been adjusted in Logistic regression models, and no significant interaction effect between milk intake and stratification factors was observed, indicating that the relationship between milk consumption and sleep disorders was independent of stratification factors. (Please refer to the Figure captions for further details (page 10, line 254-255; page 11, line 259-260))
Figure 3: Subgroup analyses of the associations (ORs, 95% CIs) between milk intake and sleep disorders among 768 participants with the regular vs. rare intake of milk. (page 10, line 252-253)
Figure 4: Subgroup analyses of the associations (ORs, 95% CIs) between milk intake and sleep disturbance among 768 participants with the regular vs. rare intake of milk. (page 11, line 257-258)
Figure caption: Logistic regression models were adjusted for age, gender, BMI, lifestyle factors and clinical factors, with exception of stratifying factors. (page 10, line 254-255; page 11, line 259-260)

Reviewer 3 Report
The authors report interesting finding from a Chinese cohort regarding milk consumption and sleep health. This population, in my knowledge, is regarded to have very low milk consumption. The manuscript is organized and results are presented appropriately in tables and figures.
Comments:
The manuscript contains English errors and needs editing.
Abstract: "Increased intake of mike (sp) was significantly associated with lower scores of sleep quality..." indicates that milk consumption was associated with poorer sleep quality. Your results state otherwise. Correct this statement.
Describe the method of participant recruitment.
Provide a "Study design" summary paragraph describing the clinic visit.
The FFQ used in the Gao study you cite contained 33 items + condiments. Describe the FFQ you used more clearly. Was the 10-item FFQ validated? What types of milk products are assessed (only fluid milk?)
That 66% of the participants (n=506) consumed milk only once per month and even "regular" consumption was >62.5 ml/week is notable. The minimal intake of milk by you participant population should be highlighted in the discussion. Can you provide a graphical representation of the distribution of milk intake? What was the highest tertile? It seems unlikely that ~60 ml per week could produce a meaningful impact on health and sleep (especially if not consumed shortly before bedtime).
Include in your discussion a comparison of the milk consumption level in relation to sleep quality in your study vs. other studies.
The manuscript contains English errors and needs editing.
Author Response
We sincerely thank you for your insightful and constructive comments, and we have improved the quality of our article according to the valuable suggestions. We would like to submit the revised version of our manuscript entitled “Associations between milk intake and sleep disorders in Chinese adults: a cross-sectional study” (ID: nutrients-2381985). The revisions in the manuscript are highlighted with yellow, and we hope our revisions will meet nutrients’ approval. Our point-by-point responses to the comments are listed below:
- The manuscript contains English errors and needs editing.
Response:
Thank you very much for your comments. We have corrected English errors in the revised version. We have actually sought the assistance of a professional language editing service to refine and proofread the English language in the manuscript.
- Abstract: "Increased intake of mike (sp) was significantly associated with lower scores of sleep quality..." indicates that milk consumption was associated with poorer sleep quality. Your results state otherwise. Correct this statement.
Response:
Thank you very much for your comment. We have modified the expression of the Abstract in revised version (page 1, line 26-29; page 1, line 36-38). Meanwhile, we have described the seven components of PSQI in the Materials and Methods: (page 4, line 134-139)
Milk intake was assessed using a food frequency questionnaire (FFQ) with 10 food items, while sleep disorders were measured using the Pittsburgh Sleep Quality Index (PSQI) with higher scores indicating poorer sleep. (page 1, line 26-29)
Increased intake of mike was significantly associated with lower scores of PSQI for sleep quality (β: −0.045, 95% CI: −0.083, −0.007) and sleep disturbances (β: −0.059, 95% CI: −0.090, −0.029), respectively. (page 1, line 36-38)
PSQI includes seven domains: sleep quality, sleep latency, sleep duration, sleep efficiency, sleep disturbances, use of sleep medications, and daytime dysfunction. The scale ranges from 0 to 3 on each domain, with higher scores indicating more severe situation. PSQI global scores are the summation of seven domain scores, and a PSQI global score of ≥ 5 indicates the presence of a sleep disorder [21, 22]. (page 4, line 134-139)
- Describe the method of participant recruitment.
Response:
Done, thank you. The details for participants’ recruitment can be found in Figure1. (page 2-3, line 85-100ï¼›page 3, line 104-106)
The study participants were enrolled at baseline within an ongoing cohort in Zhejiang Province, China. An original plan for the cohort was to recruit 3,000 general patients at baseline. The first phase plan to recruit 1,500 persons by posting posters at Wenling Hospital affiliated to Wenzhou Medical University, Zhejiang Province between March 2018 and August 2019. Ultimately, a total of 1,106 general patients aged 25–95 years residing in Taizhou Wenling City were enrolled in this study, with a response rate of 73.73%. The theoretical sample size of this study was set at 1,056 individuals to provide a specific relative precision of 2% (Type I error, 0.05; Type II error, 0.10), taking into account an anticipated 80% participation rate. Epidemiology survey data included blood-based biochemical parameters, anthropometric measurements, socio−demographics, dietary intake, and lifestyle characteristics. We excluded 338 subjects for the following reasons: i) lack of demographic and clinical disease information (n = 241); ii) lack of food frequency questionnaire or PSQI data (n = 220); iii) lack of blood samples (n = 45); iiii) have history of cow's milk proteins intolerance (n = 0), the final study sample comprised 768 participants (mean age 64.71 ± 10.51 years, male 64.32%). For detailed information on participant recruitment, please refer to Figure 1. (page 2-3, line 85-100)

Figure 1. Flowchart of study participant selection. (page 3, line 104-106)
- Provide a "Study design" summary paragraph describing the clinic visit.
Response:
Thank you for your suggestion! We have provided the "Study design" paragraph, and participant flowcharts have been added to the Figure1 (page 2, line 75-83).
The present study is a cross−sectional analysis that aims to assess the association between dietary patterns, lifestyle behaviors, and psychosomatic disorders among the population residing in the Taizhou city of Zhejiang Province, China. General patients were recruited from Wenling Hospital affiliated to Wenzhou Medical University between March 2018 and August 2019. Trained investigators completed a detailed questionnaire including items regarding socio-demographic characteristics, lifestyle variables, history of illnesses, and medication use in a face−to−face interview with the participants. Laboratory results and special examination results of the patients were collected. (page 2, line 75-83)
- The FFQ used in the Gao study you cite contained 33 items + condiments. Describe the FFQ you used more clearly. Was the 10-item FFQ validated? What types of milk products are assessed (only fluid milk?)
Response:
Thank you for your suggestion! The details for the FFQ used in our study were indeed described in Materials and Methods. (page 4, line 125-129)
Food frequency questionnaires with 10 food items were used to assess participants' average consumption frequency of foods and beverages during the previous year based on personal diet and living habits, including milk (primarily liquid cow’s milk), salt, fruits, vegetables, red meat, seafood, eggs, soy products, nuts, and sugar−sweetened beverages [17-20]. (page 4, line 125-129)
The dietary frequency questionnaire (FFQ) used in this study has been validated in several previous studies [17, 18]. The version employed in the present study is derived from Luo et al (10 item) [19] and Zhuang et al (11-item) [20].
In addition, we also conducted evaluations of the questionnaire's reliability and validity. We conducted a reliability analysis using Cronbach's alpha coefficient to evaluate the internal consistency of the questionnaire. The calculated alpha value was 0.54, which suggests a moderate level of internal consistency. For construct validity, the Kaiser-Meyer-Olkin (KMO) test showed that the KMO was 0.65 and Bartlett’s sphericity was χ 2 = 627.9, p < 0.001, indicating that the data was suitable for factor analysis. We performed factor analysis to extract four factors. According to the cumulative proportion of explained variance, these four factors accounted for a total of 55% of the total variability.
- That 66% of the participants (n=506) consumed milk only once per month and even "regular" consumption was >62.5 ml/week is notable. The minimal intake of milk by you participant population should be highlighted in the discussion. Can you provide a graphical representation of the distribution of milk intake? What was the highest tertile? It seems unlikely that ~60 ml per week could produce a meaningful impact on health and sleep (especially if not consumed shortly before bedtime).
Response:
Thank you for your suggestion! We can provide a graphical representation of the distribution of milk intake (see Supplement Figure 4) in the supplementary material. The highest tertile milk intake in the study population was >1 times/month (250 ml per month), and the median and lowest tertile were ≤1 times/month. Thus, milk intake was divided into two groups based on the distribution and median of intake: Rare (≤ 62.5 ml per week) and Regular (> 62.5 ml per week).
The point you have mentioned has been added as a limitation in the Section of the Discussion in the revised manuscript as: (page 12, line 313-318; page 13, line 359-361).
Meanwhile, in a study of food consumption by Chinese residents, it was found that the average intake of milk in normal rural areas in China was 9.1g/d. The frequency of milk intake decreased with aging [41]. The population of this study was basically middle aged and elderly population (Over 90% are older than 50 years) [42]. Therefore, the milk intake of 62.5ml/week in this study was in line with the consumption status of normal rural population. (page 12, line 313-318).
In addition, although significant associations were observed but given the low consumption of milk per week amongst 66% of our sample population, further research is warranted to confirm this finding. (page 13, line 359-361).
Include in your discussion a comparison of the milk consumption level in relation to sleep quality in your study vs. other studies.
Response:
Done, thank you for your suggestion (page 12, line 306-318)
Furthermore, our study found that individuals with a milk intake of 62.5ml or more per week had better sleep quality compared to those with a milk intake less than 62.5ml. Similar findings have been reported in previous studies. For instance, Min et al. found that individuals with a milk intake frequency of 1-2 times per week had better sleep quality compared to individuals with no milk intake per week [39]. Mozaffarian et al. found that as milk intake frequency (never, rarely, weekly, daily) increased, the probability of individuals having better sleep quality also increased [40]. Meanwhile, in a study of food consumption by Chinese residents, it was found that the average intake of milk in normal rural areas in China was 9.1g/d. The frequency of milk intake decreased with aging [41]. The population of this study was basically middle aged and elderly population (Over 90% are older than 50 years) [42]. Therefore, the milk intake of 62.5ml/week in this study was in line with the consumption status of normal rural population. (page 12, line 306-318)

Round 2
Reviewer 2 Report
I have thoroughly examined the revised edition of the article titled "Associations between milk consumption and sleep disorders in Chinese adults: a cross-sectional study" (nutrients-2381985), along with the feedback from the reviewers. I have ensured that all the recommendations have been implemented. Moreover, I have reviewed additional enhancements suggested by the other reviewers. In my opinion, this article is highly captivating and holds the potential to serve as a foundation for future investigations on the correlation between milk and sleep quality.
Author Response
We sincerely thank you for your insightful and constructive comments, and we have improved the quality of our article according to the valuable suggestions. We would like to submit the revised version of our manuscript entitled “Associations between milk intake and sleep disorders in Chinese adults: a cross-sectional study” (ID: nutrients-2381985). The revisions in the manuscript are highlighted with yellow, and we hope our revisions will meet nutrients’ approval. Our point-by-point responses to the comments are listed below:
I have thoroughly examined the revised edition of the article titled "Associations between milk consumption and sleep disorders in Chinese adults: a cross-sectional study" (nutrients-2381985), along with the feedback from the reviewers. I have ensured that all the recommendations have been implemented. Moreover, I have reviewed additional enhancements suggested by the other reviewers. In my opinion, this article is highly captivating and holds the potential to serve as a foundation for future investigations on the correlation between milk and sleep quality.
Response:
Many thanks for your positive feedback.

Reviewer 3 Report
Line 36: "Mike" needs correction.
Line 177 & following: remove the *** from statistical results reported in the text. These are used in tables and figures only.
Line 337-52: Use TG or triglycerides consistently.
Figure S4: Milk intake label needs a unit of measure.
Minor edits remain to be performed.
Author Response
We sincerely thank you for your insightful and constructive comments, and we have improved the quality of our article according to the valuable suggestions. We would like to submit the revised version of our manuscript entitled “Associations between milk intake and sleep disorders in Chinese adults: a cross-sectional study” (ID: nutrients-2381985). The revisions in the manuscript are highlighted with yellow, and we hope our revisions will meet nutrients’ approval. Our point-by-point responses to the comments are listed below:
- Line 36: "Mike" needs correction.
Response:
Thank you for your suggestion. We have corrected the error in the revised version. (page 1, line 37-39)
Increased intake of milk was significantly associated with the lower scores of PSQI for sleep quality (β: −0.045, 95% CI: −0.083, −0.007) and sleep disturbances (β: −0.059, 95% CI: −0.090, −0.029), respectively. (page 1, line 37-39)
- Line 177 & following: remove the *** from statistical results reported in the text. These are used in tables and figures only.
Response:
Thank you for presenting this point. We have removed *** in the results section of the revised version. (page 5, line 175-178)
Participants with a regular milk intake had a higher intake of fruits (< 0.001), eggs (< 0.01), nuts (< 0.01), sugar−sweetened beverages (< 0.01), less sleep disturbance (< 0.001), and lower triglycerides (0.021) compared to those with rare intake. (page 5, line 175-178)
- Line 337-52: Use TG or triglycerides consistently.
Response:
Thank you very much for your comments. We have harmonized the expression of triglycerides in the revised version. (page 13, line 336-352)
Finally, our study found that the serum triglyceride levels were negatively and linearly correlated with milk intake. The association between triglycerides and milk is a matter of debate. However, substantial evidence suggests that the whey protein in milk decreases serum triglyceride levels [50−52]. Yogurt and fermented milk reduced the triglyceride content in rats with spontaneous hypertension and might be an adjuvant to reducing hypertension and hyperlipidemia [53, 54]. Santesso et al. found that participants with higher protein intake had lower serum triglyceride levels than those with lower protein intake [55]. These results agree with our findings. Studies showed that triglycerides were associated with cardiovascular disease and sleep disorders [56]. In a longitudinal analysis, Peila et al. found that changes from restful sleep to insomnia were associated with an increased incidence of metabolic syndrome and high triglyceride levels (≥ 150 mg/dL) [57]. Shigiyama et al. demonstrated that the liver triglyceride content of mice in a chronic sleep deprivation model was significantly increased [58]. Interestingly, excessive sleep duration also increases the prevalence of metabolic syndrome and the incidence of high triglyceride levels [59]. A U−shaped relationship exists between sleep duration, metabolic syndrome, and triglyceride [60]. Nevertheless, the relationship between milk, triglyceride, and sleep requires further study. (page 13, line 336-352)
Figure S4: Milk intake label needs a unit of measure.
Response:
Thank you for your suggestion. In fact, we added the X axis to indicate this food that we have investigated is milk. The measurement unit of milk intake has been stated in the ordinate.
